# Adversarially-robust probes for Deep Networks

**Simran Ketha**[1,2]**, Nuthan Mummani** [1,2,*]**, Niranjan Rajesh** [3,†]**, Venkatakrishnan Ramaswamy**[1,2]

[1]Department of Computer Science & Information Systems,
Birla Institute of Technology & Science Pilani, Hyderabad 500078, India.
[2]Anuradha & Prashanth Palakurthi Centre for Artificial Intelligence Research,
Birla Institute of Technology & Science Pilani, Hyderabad 500078, India.
[3] Centre for Neuroscience, Indian Institute of Science, Bangalore 560012, India.
{p20200021, h20221030057, venkat}@hyderabad.bits-pilani.ac.in
niranjanrajesh02@gmail.com

## Abstract

Adversarial perturbations are strategic manipulations of input by an adversary that are aimed to cause a Deep Network to misclassify the input. Since such perturbations are employed to malicious ends, defending against them has become an important research direction. Here, we consider the question of whether the high-dimensional geometry of internal representations of Deep Networks trained with standard methods can be used to derive predictions that are robust to adversarial perturbations directed at them. To this end, we design probes on layerwise representations, whose parameters can be directly determined from the training data and/or adversarial versions thereof. We show, empirically, that such probes can have adversarial robustness that is significantly better than that of the base network, even though the probes and the base network have an identical initial substrate.

## 1 Introduction

Deep Networks have achieved extraordinary performance on many tasks and datasets. However, it is known that they suffer from unreliability of various kinds. One such type of unreliability is their susceptibility to adversarial examples, which are maliciously-crafted perturbations of inputs which are designed to elicit misclassifications by these models, while typically staying close enough to the original input in order for the perturbation to remain perceptually indistinguishable.

A number of defenses against adversarial perturbations have been proposed, with adversarial training [18] emerging as the most widely adopted approach. While effective, it remains computationally costly and tends to reduce performance on clean data [5]. Other lines of work—including methods based on data augmentation [6], regularization [20], transfer learning [12], ensembling [9], and randomized smoothing [4]—offer complementary benefits but also face their own trade-offs. Despite steady progress, building defenses that are both practical and broadly effective continues to be an open challenge [17].

Techniques to defend against adversarial perturbations typically either involve new kinds of training paradigms or expensive iterative adversarial training. Here we consider the setting where we have a pre-trained network, which hasn't been explicitly trained to be robust to adversarial perturbations. We seek to endow a degree of adversarial robustness to such networks by leveraging the high-dimensional

---

*Present affiliation: Department of Brain, Computation, and Data Science, Indian Institute of Science, Bangalore 560012, India.

†Present affiliation: Cognitive Science, University of California, San Diego, CA 92093, United States.

39th Conference on Neural Information Processing Systems (NeurIPS 2025) Workshop: Reliable ML from Unreliable Data.

geometry of their learned representations. Specifically, we build multiple classes of probes on the layerwise representations of these networks and study their adversarial robustness.

Our recent work [14] has examined the use of probes for robustness in the memorization setting (i.e. models trained with label noise). We introduce a post-hoc decoding framework, in which we build a new class of probes – the Minimum Angle Subspace Classifier (MASC). MASC constructs class-conditional subspaces from internal network representations of models which have been trained with different degrees of label noise; these subspaces are used to build a classifier. We demonstrate that such models retain latent, generalizable structure in their internal representations that enables our probe to achieve significantly better generalization than the base model, which has memorized the noisy labels. Our findings suggests that useful predictive features may persist in hidden layers but remain underutilized by standard readout mechanisms. However, the potential of leveraging a network's internal representations to defend against adversarial attacks has remained largely unexplored. We use the MASC framework to investigate the role of internal representations in the adversarial setting. Our main contributions are listed below.

- We employ MASC as a post-hoc decoding strategy to assess the capacity of internal network representations to correctly classify adversarial inputs. Specifically, we investigate whether, for adversarially perturbed test images, a model's internal representations – when decoded via MASC – produce more reliable predictions than the model's native output layer. In this setting, the class-conditional subspaces are constructed on internal representations of the clean training dataset. Our results show that, in most cases, MASC outperforms the model on adversarial data, with at least one layer consistently offering greater robustness. Indeed, this defense is attack-agnostic in the sense of not being designed to defend against a specific adversarial attack technique and does favorably on inputs perturbed using Fast Gradient Sign Method (FGSM) and Projected Gradient Descent (PGD) respectively, while retaining good accuracies on unperturbed test inputs.

- We further explore whether building class-conditional subspaces from adversarially-perturbed training data can strengthen robustness, driven by the intuition that integrating adversarial examples into subspace construction may better adapt MASC to resist such attacks. To this end, we develop *Adversarial MASC*, which constructs subspaces on adversarially-perturbed training data. We find that, while *Adversarial MASC* often surpasses standard MASC on adversarial data, but its performance deteriorates on clean test data, especially with higher $\epsilon$ values.

- Thirdly, we investigate whether constructing class-conditional subspaces from both adversarially-perturbed and clean training data can improve adversarial performance without sacrificing accuracy on clean test data. In this setting, we built an *Adversarial+ MASC*, which constructs subspaces on combined training datasets (adversarially-perturbed plus clean). Our results show that *Adversarial+ MASC* has comparable results with *Adversarial MASC* on adversarial data, while also delivering good performance on clean test data.

- Finally, we also deployed these probes on ResNet-50 pre-trained on ImageNet, where we find that they are surprisingly effective. They show over 3x improvement in adversarial accuracy when compared to the model, while having a modest drop in clean test accuracy.

Details on the models, datasets and training parameters used are available in the Appendix.

## 2 Related work

**Adversarial Attacks and Defenses**
Deep Networks' susceptibility to small, imperceptible input perturbations [24, 10] has emerged as a pressing challenge in the deep learning community and has garnered a lot of attention in recent years. The discovery of such adversarial examples quickly led to a spectrum of attack methods that evolved in both sophistication and effectiveness. Early approaches such as the Fast Gradient Sign Method (FGSM) [10] demonstrated that a single gradient-based step could reliably fool a classifier. This was soon extended to stronger iterative versions, most prominently Projected Gradient Descent (PGD) [18], which became a canonical benchmark for evaluating robustness. Optimization-based attacks, such as the Carlini–Wagner (CW) method [3], further showed that adversaries could find minimal perturbations that were highly effective at evading detection. More recently, evaluation suites like

AutoAttack [5] have emerged to standardize robustness assessment by combining multiple attacks in an adaptive and parameter-free way. Alongside white-box attacks where the model parameters are known, black-box methods have also been developed, demonstrating the surprising transferability of adversarial examples across models [21].

With the discovery of such adversarial examples, there has been an equal effort in designing defenses and other measures to minimize the effect of these attacks. The most effective, and intuitive algorithm, *Adversarial Training* [18] simply adds generated adversarial examples to the network's training diet. Although effective in dealing with adversarial examples during inference, adversarial training incurs substantial costs, including increased computational overhead and a consistent drop in clean-data performance [5]. Variants such as TRADES [27] and Free Adversarial Training [23] attempt to mitigate these trade-offs by balancing robustness with generalization or reducing computational burden. Other strategies have leveraged data augmentation [6], regularization [20], pre-training [12], ensembling [9], and distillation [22]. More recently, randomized smoothing has been proposed as a probabilistic defense that can offer certified robustness guarantees under certain perturbation regimes [4]. Despite their promise, these defenses often come with significant trade-offs, such as the need for larger training sets, multiple rounds of training, or maintaining ensembles of models, all of which increase computational and data requirements [17].

**Linear Probes in the context of Adversarial Attacks**
In the past, simple linear probes have been used to gain insight into the internal representations of intermediate layers in Deep Networks [1]. In this setup, the probes are simple linear classifiers trained iteratively on the activations of intermediate layers to minimize crossentropy loss, to assess the information available at that stage of processing in the network. Such probes are also beginning to be used in the context of adversarial examples. For instance, [9] used linear probes to fine-tune for CIFAR-10, a model that was pre-trained on ImageNet. They then used adversarial attacks that target a specific layer probe and find that doing so disrupts primarily the representations of neighboring layers, insofar as adversarial robustness is concerned. Additionally, [13] demonstrate the robustness of linear probes on models obtained by finetuning a robust pre-trained model.

## 3   Attack-agnostic probes on intermediate representations and their adversarial robustness

We ask if probes on intermediate-layer representations of Deep Networks can have better adversarial robustness than the corresponding Deep Network. Rather than use probes of the kind proposed in [1], wherein the probe weights are trained by iteratively minimizing a crossentropy classification loss, we wanted to leverage the high-dimensional geometry of class-conditional representations to create a probe, whose weights can be directly inferred from this geometry.

We use a class of probes proposed in our recent study [14] that investigated class-conditional subspaces derived from training data representations at various layers of Deep Networks, in the memorization setting. Briefly, our approach, the Minimum Angle Subspace Classifier (MASC), constructs low-dimensional class-specific subspaces by applying Principal Components Analysis (PCA) to intermediate feature representations from a chosen network layer. Given a test sample, its chosen layer representation is projected onto each class subspace, and the angle between the sample vector & its projections are computed. The predicted label corresponds to the class with the smallest such angle (i.e. highest cosine similarity).

Here, we begin by evaluating the capability of MASC constructed from clean training data. Specifically, we build MASC with class-conditioned subspaces from intermediate representations of the clean training dataset and evaluate MASC performance on adversarial test inputs. We used MASC with 1-D subspaces. For adversarially perturbing the dataset, we test this probe separately with FGSM as well as PGD40 (i.e. PGD run for 40 iterations) with different $\epsilon$ values on the test dataset. Figure 1 presents MASC results on the FGSM & PGD test dataset across different network layers for multiple models – MLPs trained on MNIST and CIFAR-10, and CNNs trained on MNIST, Fashion-MNIST, and CIFAR-10 – under standard training protocols and varying $\epsilon$ values. We have run tests on a number of $\epsilon$ budgets, some of which also correspond to cases where the perturbed images look perceptibly different. MASC is applied independently across all layers of the network.

The probes built here turn out to not be dependent on a single adversarial attack technique and we find that they perform well in the face of both the adversarial attack techniques tested here – FGSM and

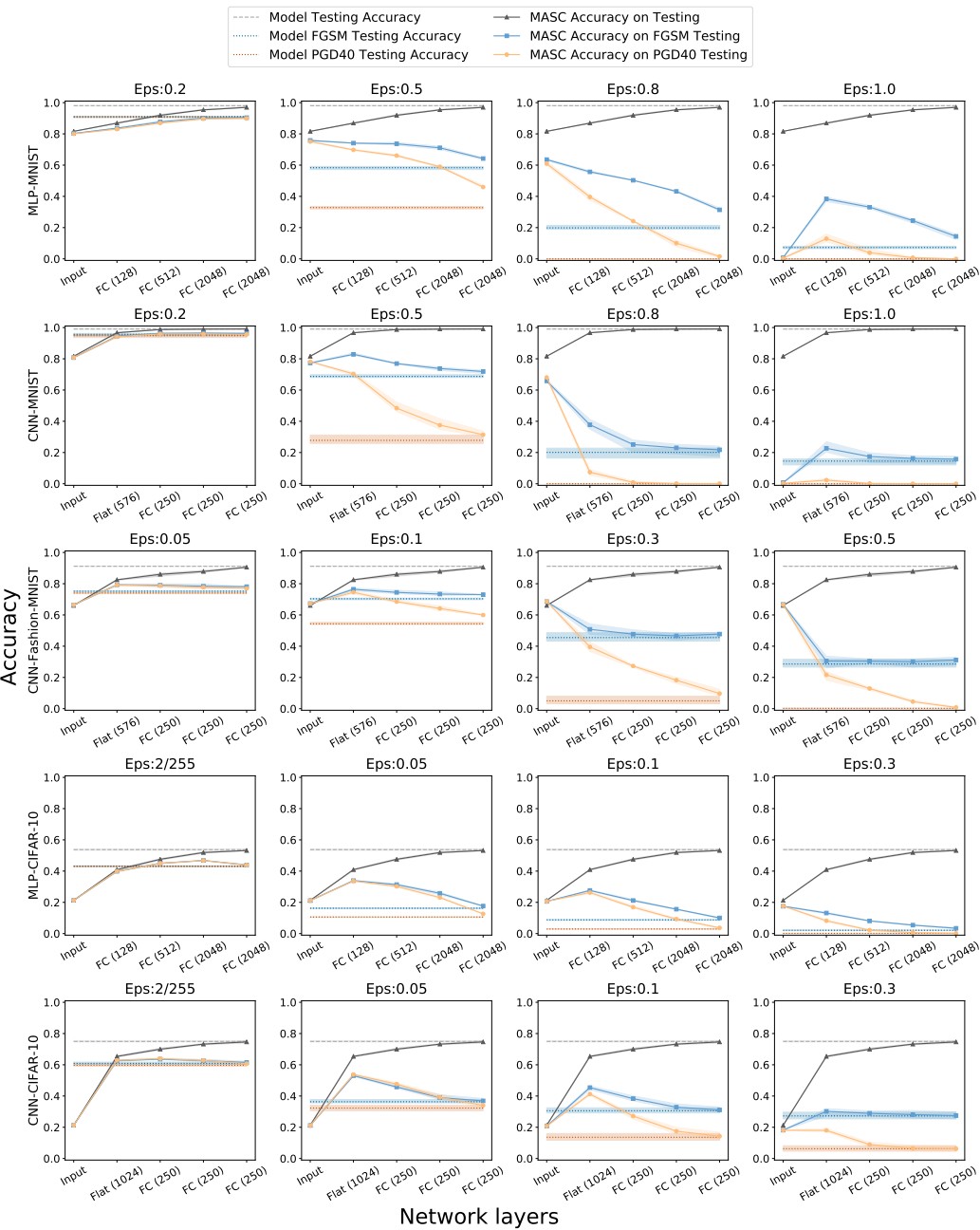

Figure 1: Minimum Angle Subspace Classifier (MASC) accuracy on adversarially perturbed test dataset and clean test dataset over the layers of the network and for varying $\epsilon$ values. Here, the data is projected onto class-specific subspaces constructed from the clean training dataset. $\epsilon$ value is presented at the top of each subplot and the rows represent model-dataset pair as indicated. For reference, the model accuracy on clean test dataset and adversarially perturbed test dataset of the corresponding model (dotted line) is also shown. PGD40 refers to Projected Gradient Descent (PGD) adversarial attacks run for 40 iterations (steps).

PGD. In many cases, for various epsilon values, the MASC accuracy on adversarial testing with these attacks is significantly better than that of the model, and this is accompanied by a modest decrease in MASC accuracy on the clean test set. Interestingly, probes on later layers tend to be less adversarially robust while having high accuracy on the clean test set. Also, in some cases, MASC applied directly on the input has good adversarial robustness, although it suffers from poorer accuracy on the test data. While we haven't run detailed comparisons with other techniques on fixed parameter values, in the Appendix, we demonstrate that the adversarial accuracy obtained here is comparable to those with other techniques for similar ranges of attack parameters. In the Appendix, we likewise run versions of our probes where the subspaces correspond to those that capture 99% of variance in the training data. Our results in this section suggest that leveraging internal representations in an attack-agnostic manner through MASC can yield more robust predictions against adversarial perturbations than the model, especially with probes on earlier layers.

## 4 Robustness of probes built on subspaces with adversarial perturbations

Here, we ask the following question: does incorporating adversarial perturbations during subspace construction enable MASC to better resist such attacks? Specifically, we investigate whether constructing class-conditional subspaces from adversarially perturbed training data – *Adversarial MASC* – improves robustness against adversarial attacks compared to subspaces derived from clean training data. Adversarial MASC's subspaces incorporate adversarial samples, which may allow it to better capture and counteract such attacks, thereby improving robustness.

Adversarial MASC results on PGD40 attack test dataset over the layers of MLP-MNIST, MLP-CIFAR-10, CNN-MNIST, CNN-Fashion-MNIST, and CNN-CIFAR-10 are shown in Figure 2 and for on FGSM attack test dataset on same models are shown in Figure 8 in the Appendix.

We find that at smaller $\epsilon$ values, MASC and Adversarial MASC perform comparably on both adversarial and clean data. However, as $\epsilon$ increases, Adversarial MASC often surpasses standard MASC on adversarial inputs, though this improvement comes at the cost of a significant drop in performance on clean test data. Also, for the case of the FGSM attack (Figure 8 of Appendix), we find in some cases that the adversarial accuracy outperforms the clean test accuracy. This is consistent with the possibility of label leaking [16] in FGSM, although we did not investigate this, in detail.

## 5 Probes built on subspaces with adversarial perturbations augmented with the clean training set samples

Using only clean training data to build subspaces preserves accuracy on clean samples to a significant extent while offering a measure of adversarial robustness. In contrast, as demonstrated in the previous section, relying solely on adversarially-perturbed data to fit subspaces improves robustness but often reduces accuracy on the clean test set significantly. Here, to explore the best of both worlds, we construct Adversarial+ MASC, which uses subspaces integrating both clean and adversarial training representations, aiming to retain clean performance while enhancing robustness. In this setting, these subspaces capture 99% of variance per class[3].

Adversarial+ MASC results on PGD40 attack test dataset over the layers of MLP-MNIST, MLP-CIFAR-10, CNN-MNIST, CNN-Fashion-MNIST, and CNN-CIFAR-10 are shown in Figure 3 and for on FGSM attack test dataset on same models are shown in Figure 10 in the Appendix. The comparative figures of Adversarial+ MASC and MASC are provided in the Appendix.

The results demonstrate that Adversarial+ MASC offers a balanced trade-off between robustness and accuracy. Specifically, while its performance on adversarial test data remains largely comparable to Adversarial MASC, it simultaneously achieves substantially higher accuracy on clean test data. For small $\epsilon$ values, Adversarial+ MASC and Adversarial MASC exhibit similar performance on both clean and adversarial test data. However, as $\epsilon$ increases, Adversarial+ MASC shows a slight

---

[3]Results corresponding to 1-D subspaces are available in the Appendix. We find that in case of 1-D subspaces, the adversarial accuracy is worse than for subspaces corresponding to those that capture 99% variance. Our hypothesis is that because the clean data points and their adversarial versions are fairly close, their distinction tends to be lost when only the first principal component is considered.

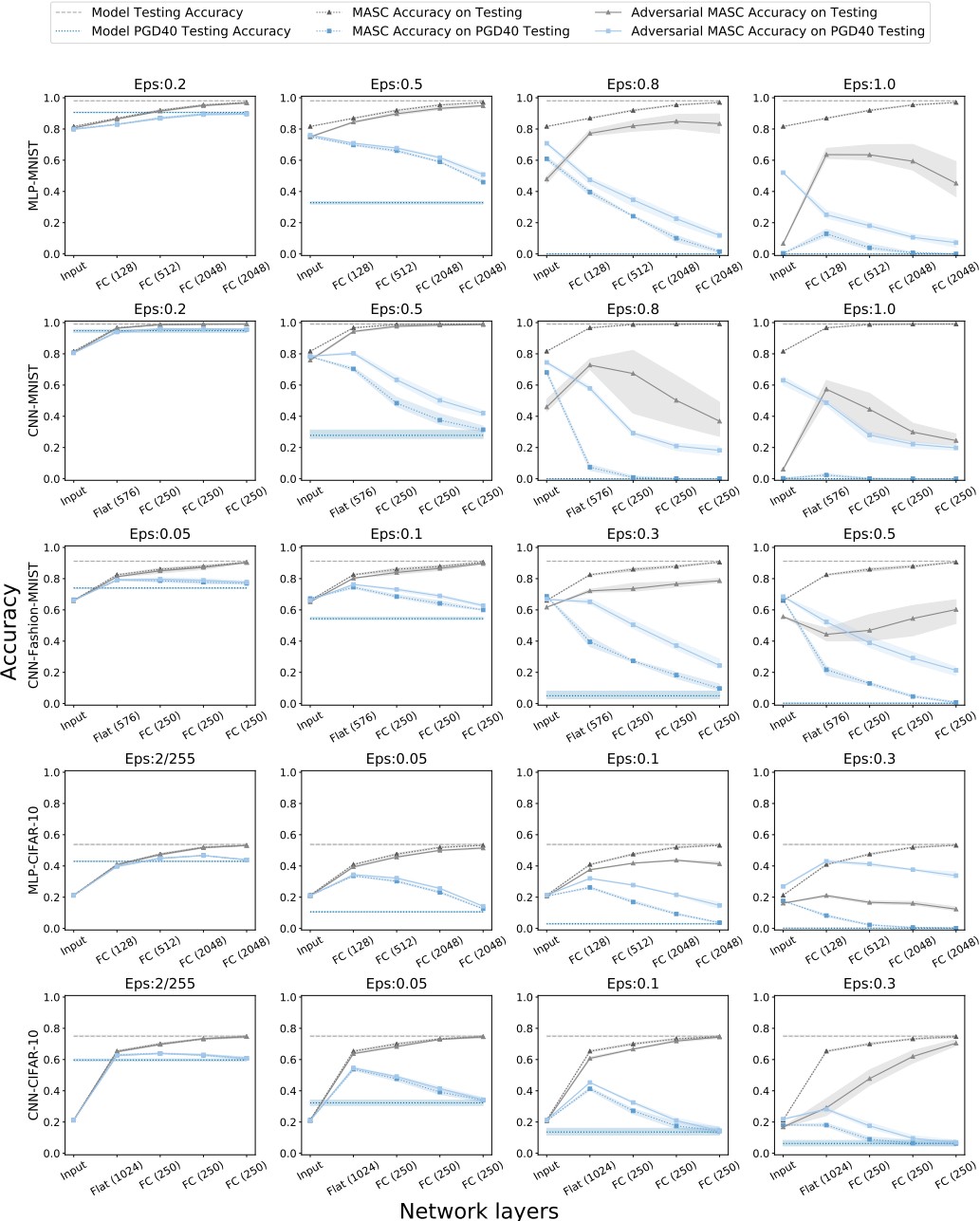

Figure 2: Adversarial Minimum Angle Subspace Classifier (Adversarial MASC) accuracy on adversarially perturbed test dataset and original test dataset over the layers of the network and for varying $\epsilon$ values. Here, the data is projected onto class-specific subspaces constructed from PGD40 training dataset. $\epsilon$ value is presented at the top of each plot and the columns represent model-dataset pair as indicated. For reference, the model accuracy on original test dataset and adversarially perturbed test dataset of the corresponding model (dotted line) is also shown. MASC accuracy on testing (dotted line) and PGD40 testing (dotted line) when data is projected onto original training subspaces is overlaid for comparison. PGD40 refers to Projected Gradient Descent (PGD) adversarial attacks run for 40 iterations (steps).

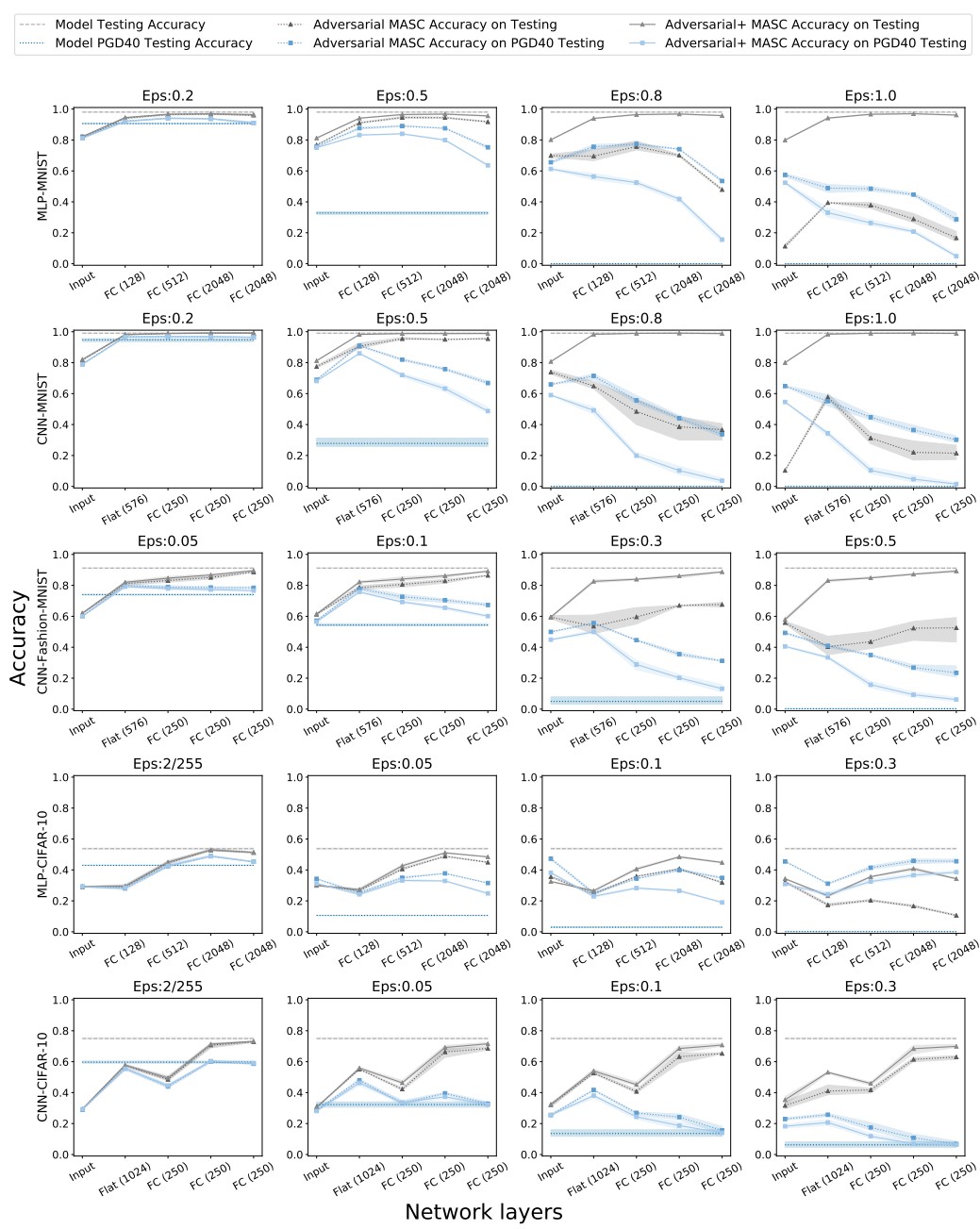

Figure 3: Adversarial+ Minimum Angle Subspace Classifier (Adversarial+ MASC) accuracy on adversarially perturbed test dataset and clean test dataset over the layers of the network and for varying $\epsilon$ values. Here, the data is projected onto class-specific subspaces constructed from PGD40 and clean training dataset. $\epsilon$ value is presented at the top of each plot and the columns represent model-dataset pair as indicated. For reference, the model accuracy on clean test dataset and adversarially perturbed test dataset of the corresponding model (dotted line) is also shown. Adversarial MASC accuracy on testing (dotted line) and PGD40 testing (dotted line) when data is projected onto subspaces corresponding to only adversarial data is overlaid for comparison.

reduction in adversarial robustness compared to Adversarial MASC, but maintains significantly stronger accuracy on clean test data.

Adversarial+ MASC can be considered somewhat analogous to Adversarial Training [18, 27] due to the inclusion of both clean and adversarial samples in the model training diet. Although, it is worth pointing out that traditional Adversarial Training involves expensive iterations that train a full Deep Network on the additional samples whereas our Adversarial+ MASC only needs a single forward propagation of these adversarial examples to construct the class-conditional subspaces. Despite this, we see comparable performance between the two robustness regimes in Table 2 of the Appendix.

## 6 Probes on ResNet-50 pre-trained on ImageNet

We also ran a smaller scale of experiments[4] on a ResNet-50 model pre-trained on the ImageNet dataset with 1000 classes, which is the largest model/dataset we have tested. We ran experiments only for PGD with 10 iterations and with a single $\epsilon$ budget (0.3). For this value of $\epsilon$, the adversarial perturbations look perceptually indistinguishable from the original images. We ran the previously discussed MASC variants on three of the layerwise outputs. See Table 1 for details of the clean test accuracies and adversarial test accuracies we obtained for these probes, along with the corresponding baseline accuracies obtained on the ResNet-50 model.

Table 1: MASC, Adversarial MASC and Adversarial+ MASC accuracies on the ResNet-50 model. While the Testing column refers to accuracy on the ImageNet validation dataset, PGD Testing column refers to accuracy on an adversarial version of the same (adversarially attacked using PGD; $\epsilon$-value: 0.3; iterations: 10). MASC either uses 600(c, a, a+) or 300(a) images per class in fitting subspaces, while 'c', 'a' and 'a+' refers to clean, adversarial and adversarial+(50% clean + 50% adversarial) training images respectively. We used probes on the outputs of three different layers of the ResNet-50 model namely avg_pool, conv5_block3_out, conv5_block2_out. During subspace construction of MASC, 1-D subspaces were used per class for probes in all three layers and an additional experiment was conducted on the avg_pool layer with subspaces each capturing 99% variance of the class-conditioned training data.

| ResNet-50 layer & nature of subspaces | MASC trained on | Testing | PGD Testing |
|---|---|---|---|
| avg_pool (2048 dimensions) 1-D subspaces | 600c | 0.51 | 0.26 |
| | 300a | 0.52 | 0.32 |
| | 600a | 0.53 | 0.33 |
| | 600a+ | 0.55 | 0.30 |
| conv5_block3_out (100352 dimensions) 1-D subspaces | 600c | 0.49 | 0.30 |
| | 300a | 0.49 | 0.33 |
| | 600a | 0.51 | 0.34 |
| | 600a+ | 0.52 | 0.32 |
| conv5_block2_out (100352 dimensions) 1-D subspaces | 600c | 0.33 | 0.25 |
| | 300a | 0.34 | 0.28 |
| | 600a | 0.34 | 0.29 |
| | 600a+ | 0.36 | 0.28 |
| avg_pool (2048 dimensions) Subspaces capturing 99% variance | 600c | 0.53 | 0.28 |
| | 300a | 0.57 | 0.45 |
| | **600a** | **0.59** | **0.45** |
| | 600a+ | 0.59 | 0.43 |
| ResNet-50 Model accuracies | – | 0.65 | 0.13 |

We find that the probes can be remarkably effective in this case. In particular, an Adversarial MASC probe trained on the avg_pool layer using class-conditional subspaces that capture 99% variance results in 45% adversarial accuracy, which is 346% better than the adversarial accuracy of 13% obtained with the model. This probe comes with a clean test accuracy of 59% which is marginally lower than the clean test accuracy of 65% obtained with the model.

---

[4]See Appendix for details of experimental set-up.

We also sought to examine the dependence of the probe performance on the ambient dimensionality of the layerwise representations. Two of the layers tested have about 50x the ambient dimensionality of the avg_pool layer. For MASC variants that use 1-D subspaces, we don't find that higher ambient dimensionality results in better performance. Likewise, we considered how the performance of the MASC probes is dependent on the number of images per class used to fit subspaces. For MASC variants that use 1-D subspaces, we found that doubling the number of images per class from 300 to 600 results only in marginal gains in acccuracy.

# 7   Discussion

Here, we considered the setting where we have Deep Networks pre-trained on a dataset via standard methods that do not include any adversarial training. Our goal was to ask whether we can leverage existing high-dimensional layerwise representations to obtain some measure of adversarial robustness, without resorting to expensive iterative (re)training methods. We find indeed that simple computationally-inexpensive probes, which are not even specifically designed for the adversarial setting, can already offer a modest degree of adversarial robustness without significant sacrifices on clean test accuracy. Secondly, we built variants of these probes that used adversarial perturbations of the training data. We find that these probes can have better adversarial robustness than the previous class of probes, especially for larger $\epsilon$ budgets, although this comes at the cost of clean test accuracy. With a view to have improved adversarial test accuracy, as obtained in the previous case, but without significantly impacting clean test accuracy, we also created versions of probes that fit subspaces to clean training data augmented with adversarially perturbed versions of the training data. Here, we find that the probes can have somewhat better adversarial accuracy than our attack-agnostic probes, while taking only a modest loss in clean test accuracy. We also tested these probes on ResNet-50 trained on ImageNet – our largest model/dataset. Here we find that the probes are particularly effective. Indeed, our probes are able to outperform the model's adversarial accuracy by up to 346%, while only having a slightly worse clean test accuracy.

The work in its present form carries some limitations. On the one hand, we have not run detailed comparisons with other techniques for identical values of parameters. However, we show (in the Appendix) that our results are comparable to those from other techniques for similar parameter values, with our techniques often requiring less computational overhead. For some models, we haven't run probes on all layers; it is possible that some of these layers indeed show better performance. It would also be interesting to see how the probes perform on other attack techniques such as AutoAttack and Carlini-Wagner.

The use of probes on layerwise representations of pre-trained networks towards obtaining better adversarial robustness is a research direction that hasn't yet received adequate attention. Indeed, it is somewhat surprising that existing representations learned by networks trained with standard methods carry a significant measure of adversarial robustness that is underutilized by the networks. Our work shows the initial promise of using probes to take advantage of robustness present in these representations. The detailed mechanisms that underlie their effectiveness are poorly understood and merit further investigation. For example, the three variants of probes designed here do not show significant differences in accuracy for smaller $\epsilon$ budgets, the reasons for which are unclear. It is possible that a detailed understanding of the mechanisms here could not only serve to improve such probes, but also lead to better adversarial training methods that better leverage this latent adversarial robustness.

**Acknowledgments**

Simran Ketha was supported by an APPCAIR Fellowship, from the Anuradha & Prashanth Palakurthi Centre for Artificial Intelligence Research (APPCAIR). Nuthan Mummani was supported in part by a Research Assistantship from APPCAIR. The work was supported in part by an Additional Competitive Research Grant from BITS to Venkatakrishnan Ramaswamy. The authors acknowledge the computing time provided on the High Performance Computing facility, Sharanga, at the Birla Institute of Technology and Science - Pilani, Hyderabad Campus.

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

# A  Model and training details

We conduct experiments using Multi-Layer Perceptrons (MLPs) trained on MNIST [8] and CIFAR-10 [15]; Convolutional Neural Networks (CNNs) trained on MNIST, Fashion-MNIST [25], and CIFAR-10 and ResNet50 [11] pre-trained on ImageNet[7].

The MLP model consists of four hidden layers with 128, 512, 2048, and 2048 units, respectively. Each hidden layer is followed by a $ReLU$ activation, while a $softmax$ activation is applied at the output for classification. Training was performed using the SGD optimizer with a learning rate of $1 \times 10^{-3}$ and momentum of 0.9. A batch size of 32 was used across all experiments. Input data was normalized by dividing pixel values by 255.

The CNN model is composed of three convolutional blocks, each containing two convolutional layers followed by a max pooling layer. The convolutional layers use 16, 32, and 64 filters, respectively, with a stride of 1 and a kernel size of $3 \times 3$. The max pooling layers use a stride of 1 and a kernel size of $2 \times 2$. These blocks are followed by three fully connected layers, each with 250 units. The model was trained with the Adam optimizer using a learning rate of $2 \times 10^{-4}$. For MNIST and Fashion-MNIST, a batch size of 32 was used, while for CIFAR-10 the batch size was 128. Input data was normalized by subtracting the mean and dividing by the standard deviation for each channel. ReLU activations were applied after all layers except pooling, and a softmax activation was used at the final classification layer.

We have used ResNet-50 pre-trained on ImageNet. For pre-processing the images, we first applied zero-padded to resize each image to 224 x 224 and then performed the standard pre-processing step for ResNet-50 using TensorFlow library. Pre-proessed images were used in all experiments in this work. Layerwise outputs of these inputs are flattened and then used in subspace construction.

In our adversarial experiments with MNIST, Fashion-MNIST, and CIFAR-10, we evaluated robustness under FGSM and PGD attacks (L-inf) using the following dataset-specific $\epsilon$ values: MNIST (0.2, 0.5, 0.8, 1.0), Fashion-MNIST (0.05, 0.1, 0.3, 0.5), and CIFAR-10 (2/255, 0.05, 0.1, 0.3). For PGD attacks, we performed 40 iterations. On the ImageNet dataset, we applied an $\epsilon$=0.3 with 10 PGD iterations. The attacks were generated by algorithms adapted from [10] and [18].

For the MLP, experiments were conducted on all layers of the network, whereas for the CNN, they were restricted to the last four layers. For ResNet-50, the experiments were conducted on avg_pool, conv5_block3_out and conv_5block2_out layers. For MASC algorithms, refer to [14].

In the plots, Model Testing Accuracy refers to models' accuracy on the clean test dataset, Model FGSM Testing Accuracy to models' accuracy on an FGSM-attacked test dataset, MASC Accuracy on Testing to MASC performance on the original test dataset, and MASC Accuracy on FGSM Testing to MASC performance on the FGSM-perturbed test dataset. Similar understading Accuracy was used as the primary evaluation metric throughout. Results are averaged over three independent training runs, with shaded regions in the plots indicating the range across runs, except for ResNet50, where results are reported from a single run since it is a pre-trained network.

Experiments were performed on workstation having NVIDIA GeForce RTX 3090s and server equipped with Tesla A100 and Tesla H100 GPUs. The workstation operated on Ubuntu 20.04.3 LTS and the server on Rocky Linux 8.10 (Green Obsidian). All MLP-CNN models were implemented in Python using the PyTorch library and ResNet-50 was implemented using Keras/TensorFlow. Memory usage varied across experiments depending on the model and dataset. For reproducibility, we set torch.manual_seed to 42 in case of CNN and MLP models. Most experiments completed within 12–24 hours.

# B  Comparing MASC with other Adversarial Defenses

We compared our implementations of MASC with several widely-studied and state-of-the-art defense measures on CIFAR-10 (Table 2) and ImageNet (Table 3). The reported accuracy results are taken directly from the original papers under PGD [18] attacks. For both datasets, we include three variants of our MASC model (Standard, Adversarial, and Adversarial+) and representative defenses from recent adversarial robustness literature. These include methods that augment training with adversarial or synthetic examples [18, 27, 2] and defenses that employ specialized regularization techniques [19, 26].

Table 2: Comparison of different defense methods against adversarial attacks on CIFAR-10. Our results reported are averaged over three runs for the best layer.

| Defense Method | Architecture | $\epsilon$-value | Clean Acc. | Adv. Acc. |
|---|---|---|---|---|
| MASC (Ours) | CNN | 0.05 (12.75/255) | 65.35 | 53.81 |
| Adversarial MASC (Ours) | CNN | 0.05 (12.75/255) | 74.57 | 54.65 |
| Adversarial+ MASC (Ours) | CNN | 0.05 (12.75/255) | 71.58 | 46.10 |
| MixUp Inference [19] | ResNet50 | 8/255 | 82.90 | 31.00 |
| Standard AT [18] | ResNet50 | 8/255 | 87.30 | 47.04 |
| TRADES AT [27] | ResNet18 | 8/255 | 84.92 | 56.61 |

Table 3: Comparison of different defense methods against adversarial attacks on ImageNet.

| Defense Method | Architecture | $\epsilon$-value | Clean Acc. | Adv. Acc. |
|---|---|---|---|---|
| MASC (Ours) | ResNet50 | 0.3 (76.5/255) | 53.00 | 28.00 |
| Adversarial MASC (Ours) | ResNet50 | 0.3 (76.5/255) | 59.00 | 45.00 |
| Adversarial+ MASC (Ours) | ResNet50 | 0.3 (76.5/255) | 59.00 | 43.00 |
| Standard AT [18] | ResNet50 | 4/255 | 62.42 | 33.58 |
| Augmentation Warmup [2] | DeIT-S | 4/255 | 66.62 | 36.56 |
| MIMIR [26] | ViT-B | 4/255 | 76.98 | 53.84 |

Broadly, we find that our results are in the ballpark of results obtained via other techniques, while, in many cases, requiring significantly smaller computational overhead.

## C Additional results with PGD attack

In this section, we present additional results with Projected Gradient Descent (PGD) attack. For subspaces corresponding to 99% variance captured, MASC and Adversarial MASC results are shown in Figure 4. Adversarial+ MASC results with only top one principal component are shown in Figure 5. Results comparing Adversarial+ MASC with MASC for only top principal component is shown in Figure 6 and for 99% variance captured in Figure 7.

## D Additional results on with FGSM attack

In this section, we present the results with Fast Gradient Sign Method (FGSM) attack. Here, we show results with probes subspaces using only top one principle component and 99% variance explained. MASC and Adversarial MASC results with subspaces corresponding only top one principle component and 99% variance captured per class are shown in Figure 8 and Figure 9 respectively.

Adversarial+ MASC results with subspaces corresponding to only one top principal component and 99% variance captured per class are shown in Figure 10 and 11 respectively. Results comparing Adversarial+ MASC with MASC for only top principal component is shown in Figure 12 and for 99% variance captured in Figure 13 respectively.

## E Additional results with MASC

Here, we present additional results with both the attack for MASC using subspaces corresponding to 99% variance captured. Figure 14 presents MASC results on the FGSM & PGD test dataset across different network layers for multiple models under standard training protocols and varying $\epsilon$ values.

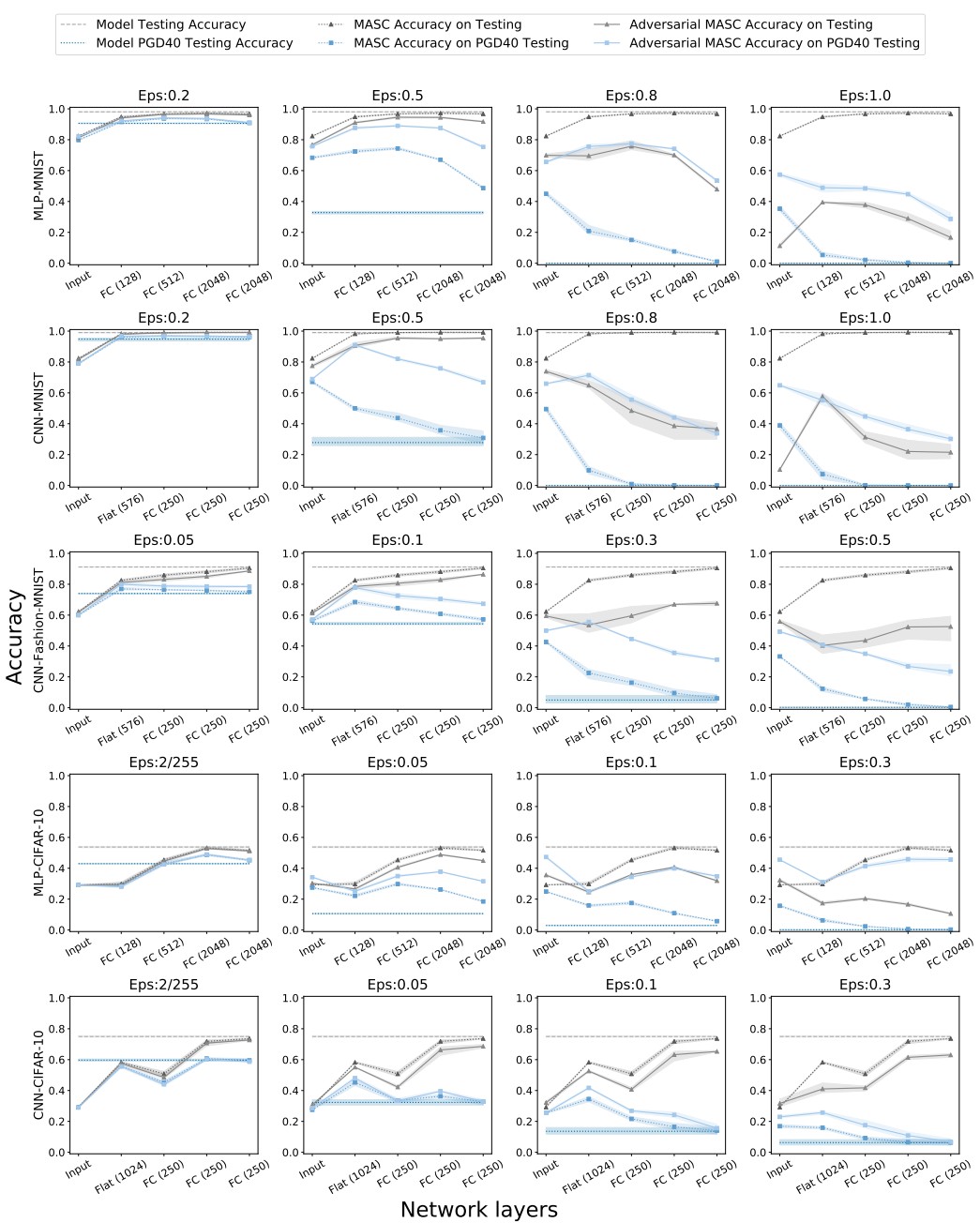

Figure 4: Minimum Angle Subspace Classifier (MASC) accuracy and Adversarial Minimum Angle Subspace Classifier (Adversarial MASC) accuracy on adversarially perturbed test dataset and clean test dataset over the layers of the network and for varying $\epsilon$ values. For MASC, the data is projected onto class-specific subspaces constructed from clean training dataset and for Adversarial MASC, the data is projected onto class-specific subspaces constructed from PGD40 training dataset. $\epsilon$ value is presented at the top of each plot and the columns represent model-dataset pair as indicated. For reference, the model accuracy on clean test dataset and adversarially perturbed test dataset of the corresponding model (dotted line) is also shown. PGD40 refers to Projected Gradient Descent (PGD) adversarial attacks run for 40 iterations (steps).

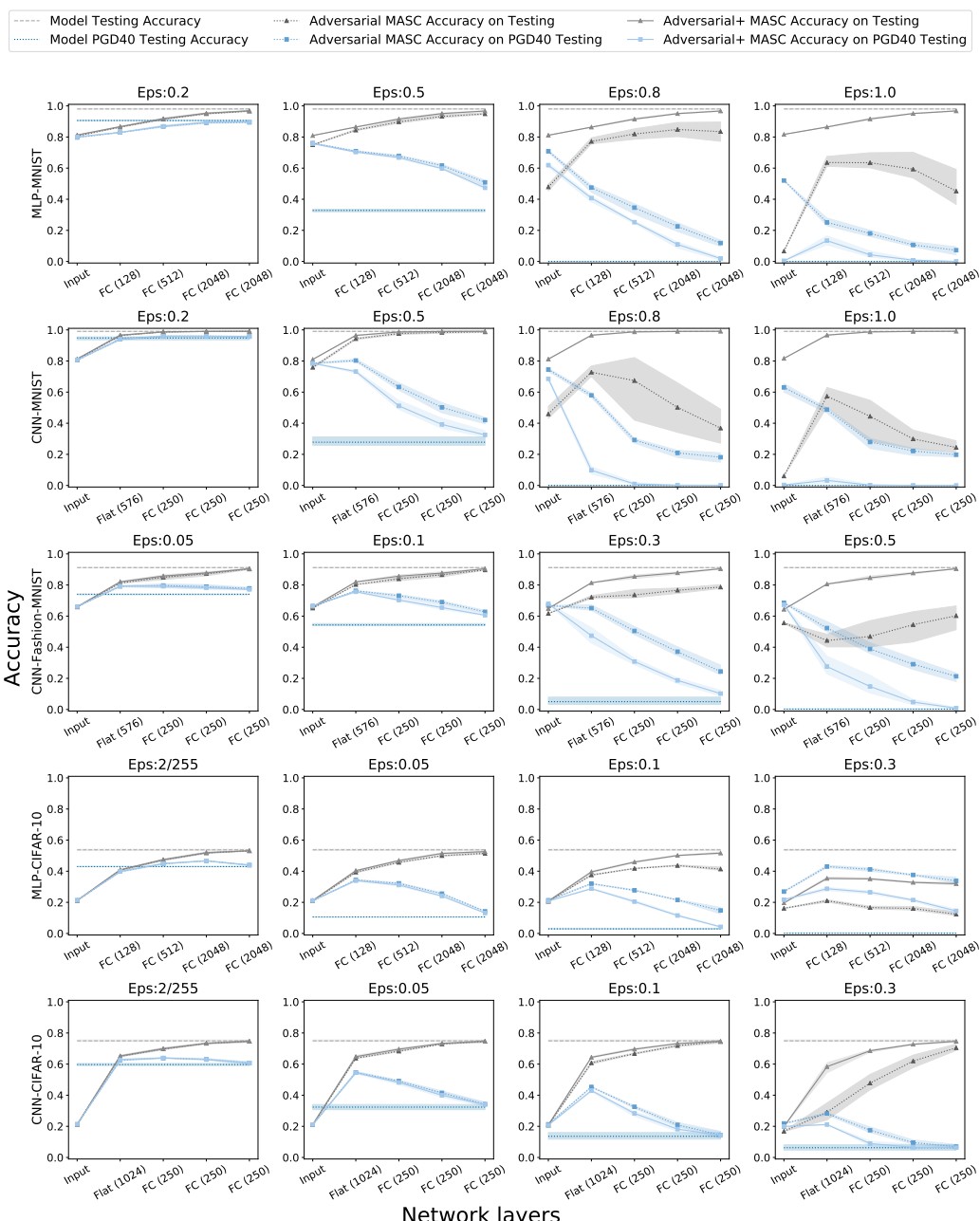

Figure 5: Adversarial+ Minimum Angle Subspace Classifier (Adversarial+ MASC) accuracy on adversarially perturbed test dataset and clean test dataset over the layers of the network and for varying $\epsilon$ values. Here, the data is projected onto class-specific subspaces constructed from PGD40 and clean training dataset. $\epsilon$ value is presented at the top of each plot and the columns represent model-dataset pair as indicated. For reference, the model accuracy on clean test dataset and adversarially perturbed test dataset of the corresponding model (dotted line) is also shown. Adversarial MASC accuracy on testing (dotted line) and PGD40 testing (dotted line) when data is projected onto subspaces corresponding to only adversarial data is overlaid for comparison.

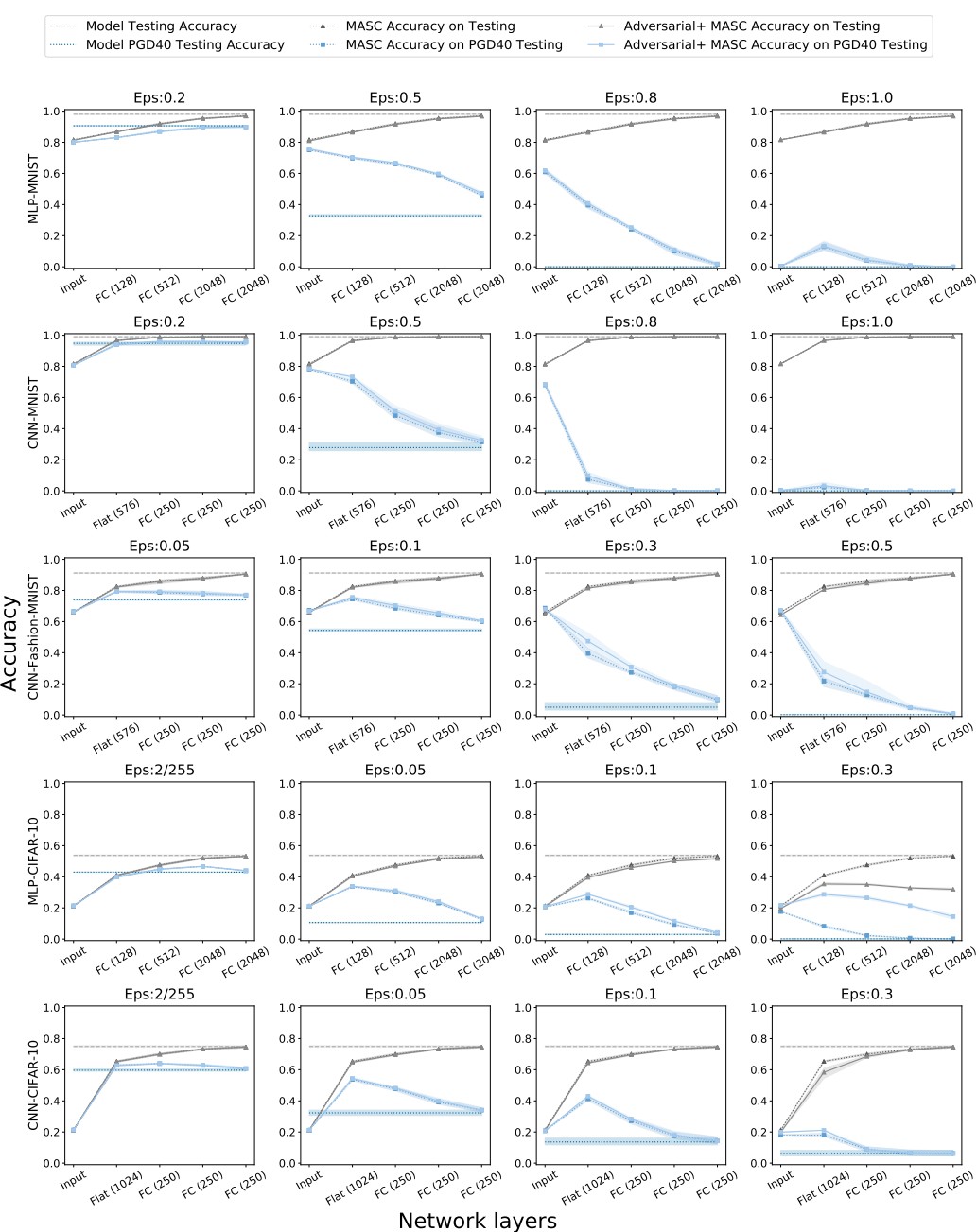

Figure 6: Adversarial+ Minimum Angle Subspace Classifier (Adversarial+ MASC) accuracy on adversarially perturbed test dataset and clean test dataset over the layers of the network and for varying $\epsilon$ values. Here, the data is projected onto class-specific subspaces constructed from PGD40 and clean training dataset. $\epsilon$ value is presented at the top of each plot and the columns represent model-dataset pair as indicated. For reference, the model accuracy on clean test dataset and adversarially perturbed test dataset of the corresponding model (dotted line) is also shown. MASC accuracy on testing (dotted line) and PGD40 testing (dotted line) when data is projected onto subspaces corresponding to only clean data is overlaid for comparison.

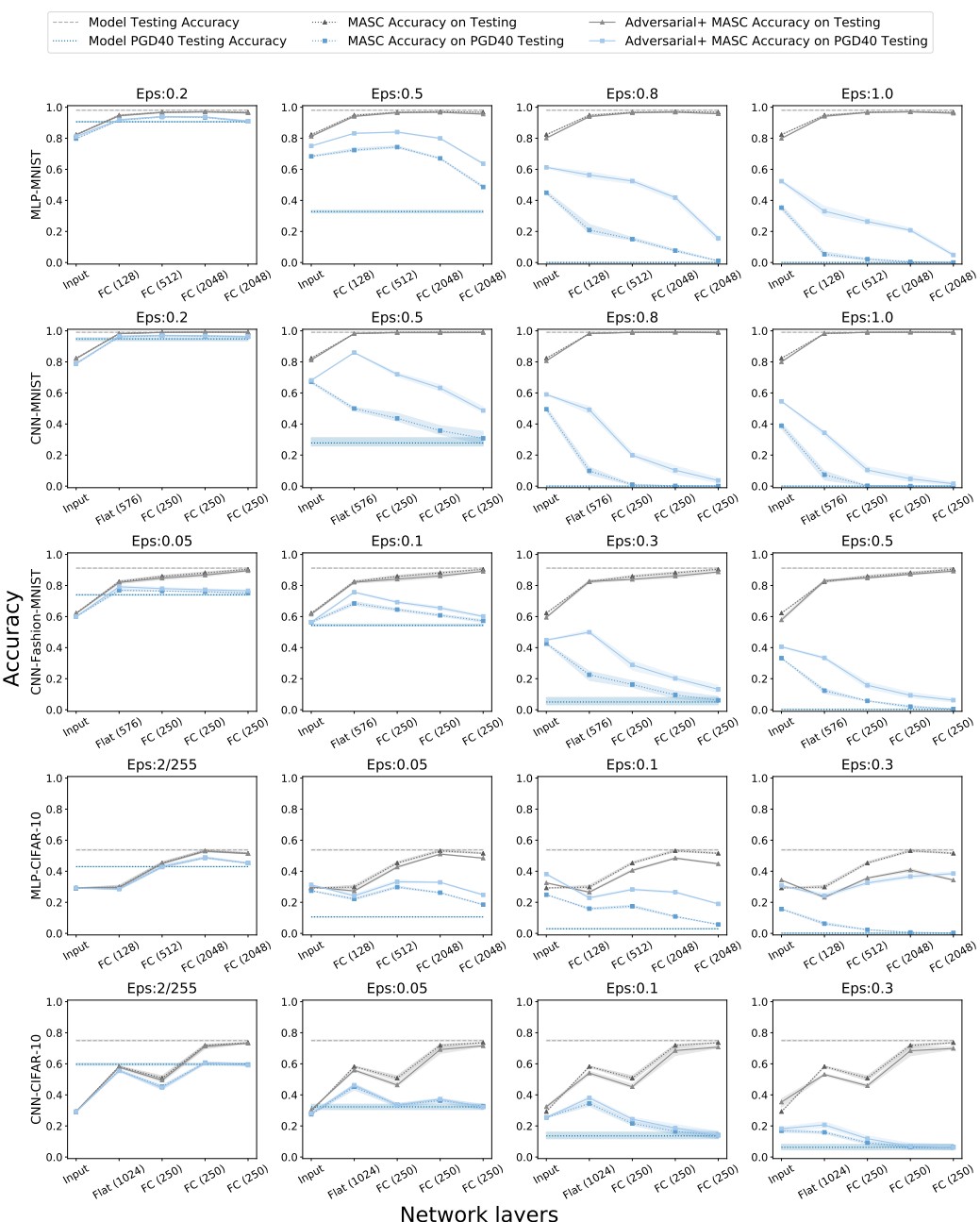

Figure 7: Adversarial+ Minimum Angle Subspace Classifier (Adversarial+ MASC) accuracy on adversarially perturbed test dataset and clean test dataset over the layers of the network and for varying $\epsilon$ values. Here, the data is projected onto class-specific subspaces constructed from PGD40 and clean training dataset. $\epsilon$ value is presented at the top of each plot and the columns represent model-dataset pair as indicated. For reference, the model accuracy on clean test dataset and adversarially perturbed test dataset of the corresponding model (dotted line) is also shown. MASC accuracy on testing (dotted line) and PGD40 testing (dotted line) when data is projected onto subspaces corresponding to only clean data is overlaid for comparison.

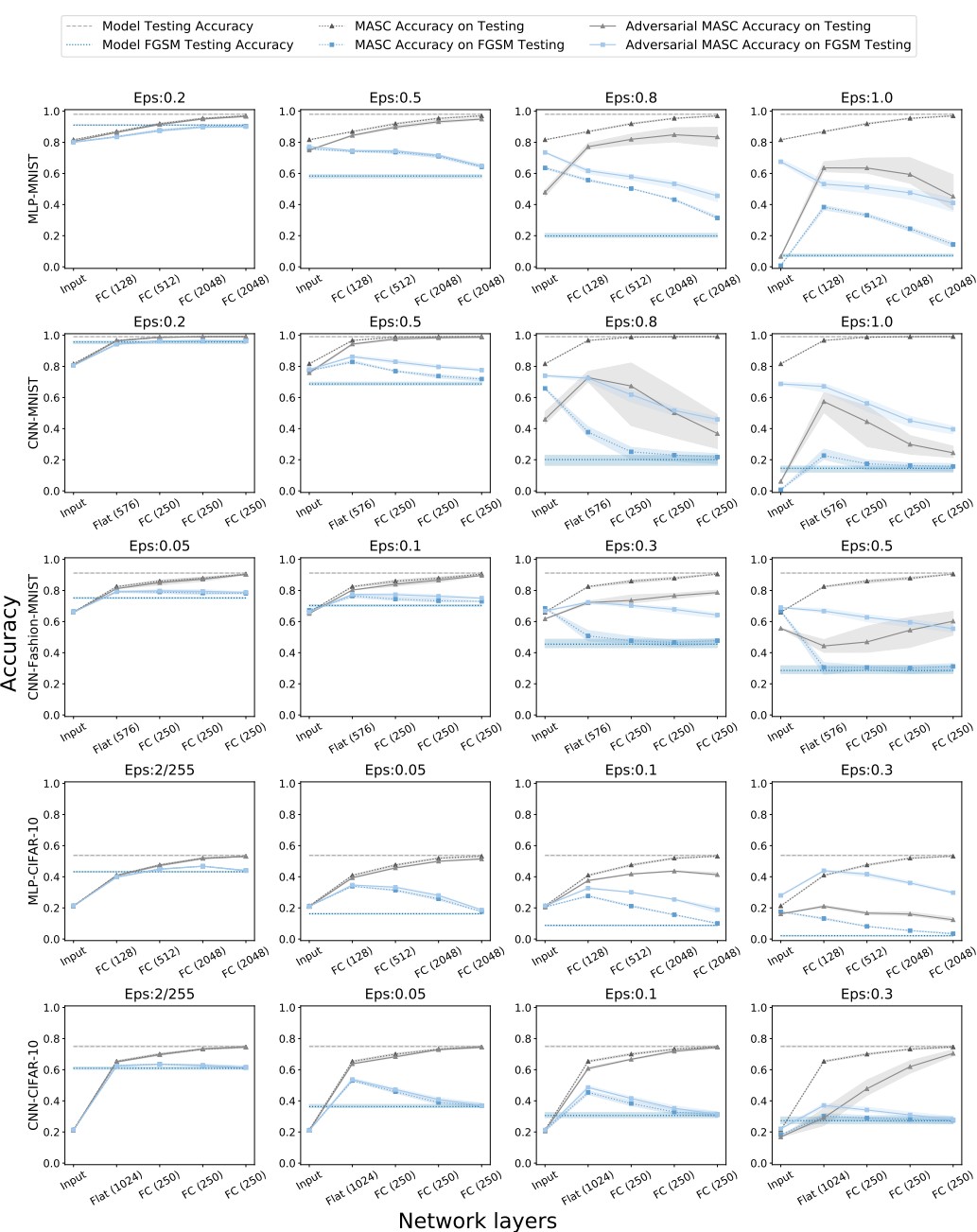

Figure 8: Adversarial Minimum Angle Subspace Classifier (Adversarial MASC) accuracy on adversarially perturbed test dataset and original test dataset over the layers of the network and for varying $\epsilon$ values. Here, the data is projected onto class-specific subspaces constructed from FGSM training dataset. $\epsilon$ value is presented at the top of each plot and the columns represent model-dataset pair as indicated. For reference, the model accuracy on original test dataset and adversarially perturbed test dataset of the corresponding model (dotted line) is also shown. MASC accuracy on testing (dotted line) and FGSM testing (dotted line) when data is projected onto original training subspaces is overlaid for comparison.

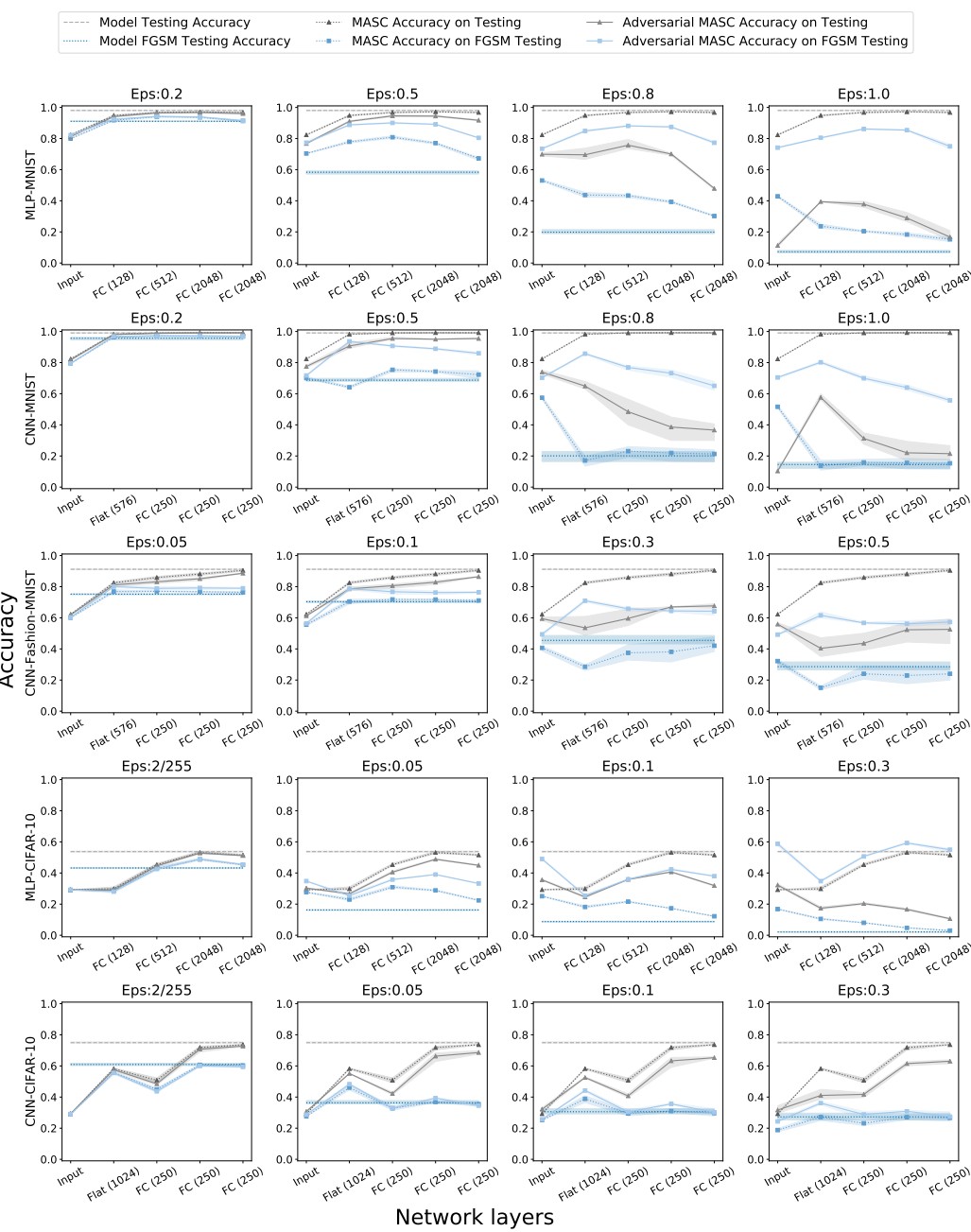

Figure 9: Minimum Angle Subspace Classifier (MASC) accuracy and Adversarial Minimum Angle Subspace Classifier (Adversarial MASC) accuracy on adversarially perturbed test dataset and clean test dataset over the layers of the network and for varying $\epsilon$ values. For MASC, the data is projected onto class-specific subspaces constructed from clean training dataset and for Adversarial MASC, the data is projected onto class-specific subspaces constructed from FGSM training dataset. $\epsilon$ value is presented at the top of each plot and the columns represent model-dataset pair as indicated. For reference, the model accuracy on clean test dataset and adversarially perturbed test dataset of the corresponding model (dotted line) is also shown.

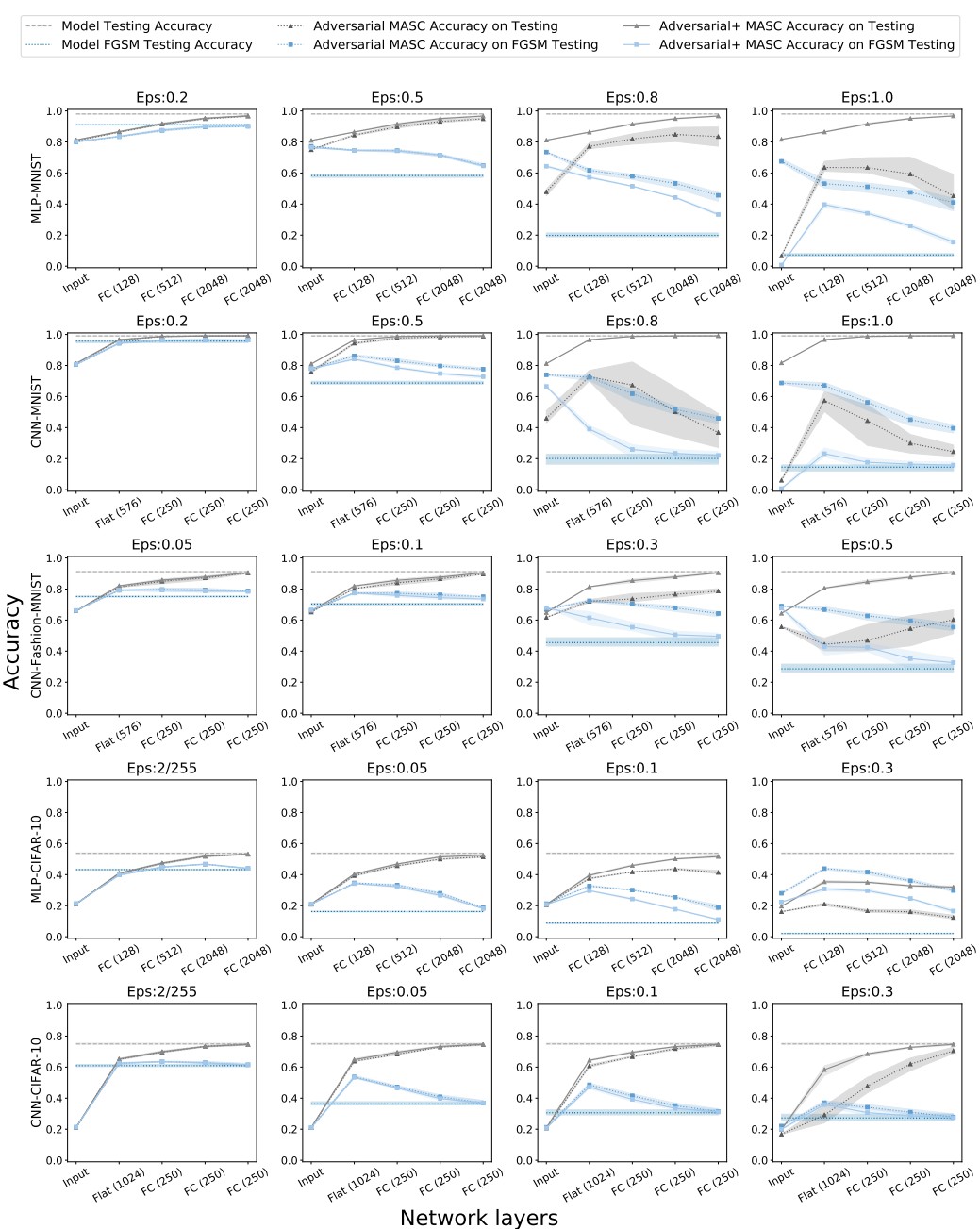

Figure 10: Adversarial+ Minimum Angle Subspace Classifier (Adversarial+ MASC) accuracy on adversarially perturbed test dataset and clean test dataset over the layers of the network and for varying $\epsilon$ values. Here, the data is projected onto class-specific subspaces constructed from FGSM and clean training dataset. $\epsilon$ value is presented at the top of each plot and the columns represent model-dataset pair as indicated. For reference, the model accuracy on clean test dataset and adversarially perturbed test dataset of the corresponding model (dotted line) is also shown. Adversarial MASC accuracy on testing (dotted line) and FGSM testing (dotted line) when data is projected onto subspaces corresponding to only adversarial data is overlaid for comparison.

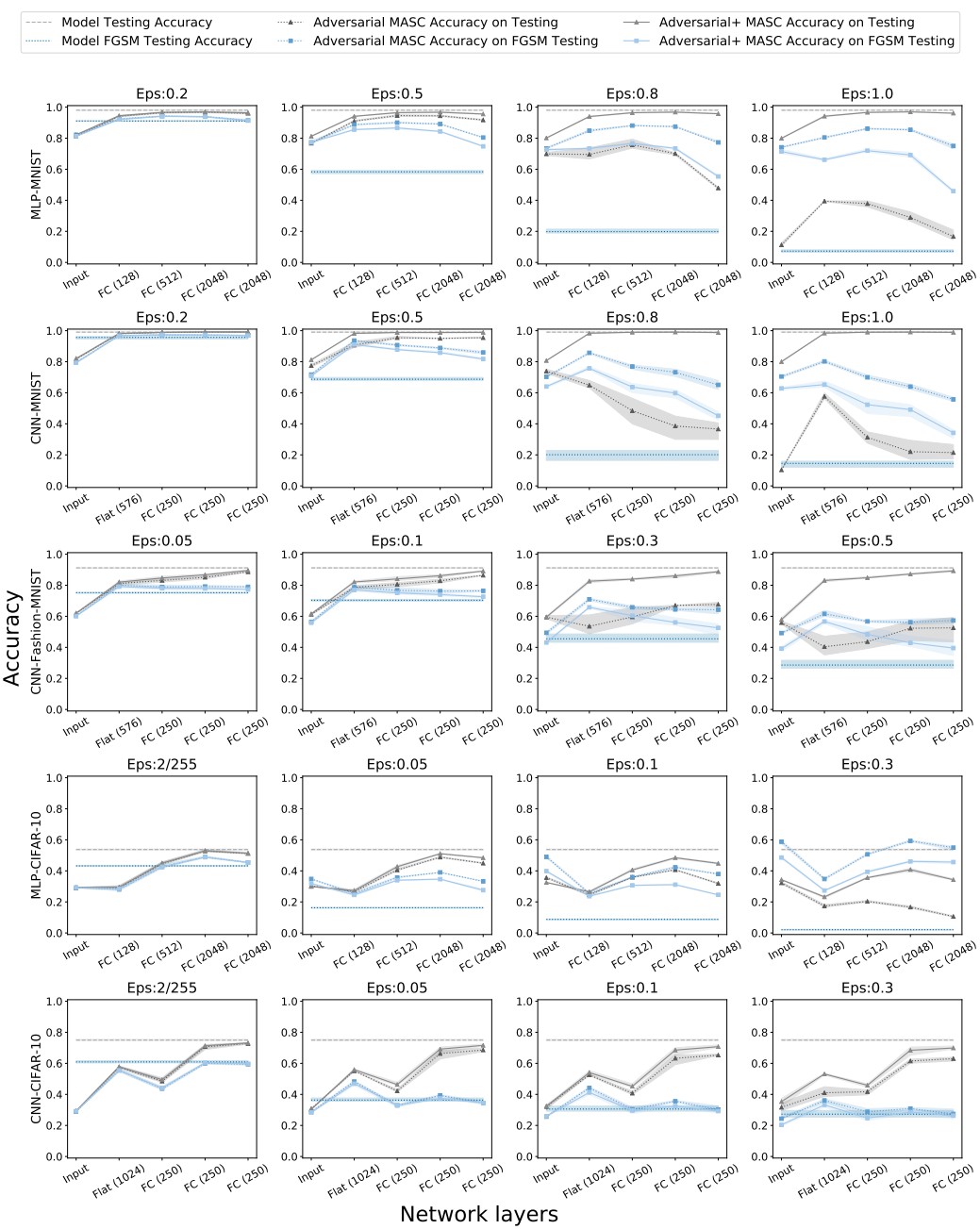

Figure 11: Adversarial+ Minimum Angle Subspace Classifier (Adversarial+ MASC) accuracy on adversarially perturbed test dataset and original test dataset over the layers of the network and for varying $\epsilon$ values. Here, the data is projected onto class-specific subspaces constructed from FGSM and clean training dataset. $\epsilon$ value is presented at the top of each plot and the columns represent model-dataset pair as indicated. For reference, the model accuracy on original test dataset and adversarially perturbed test dataset of the corresponding model (dotted line) is also shown. Adversarial-MASC accuracy on testing (dotted line) and FGSM testing (dotted line) when data is projected onto subspaces corresponding to only adversarial data is overlaid for comparison.

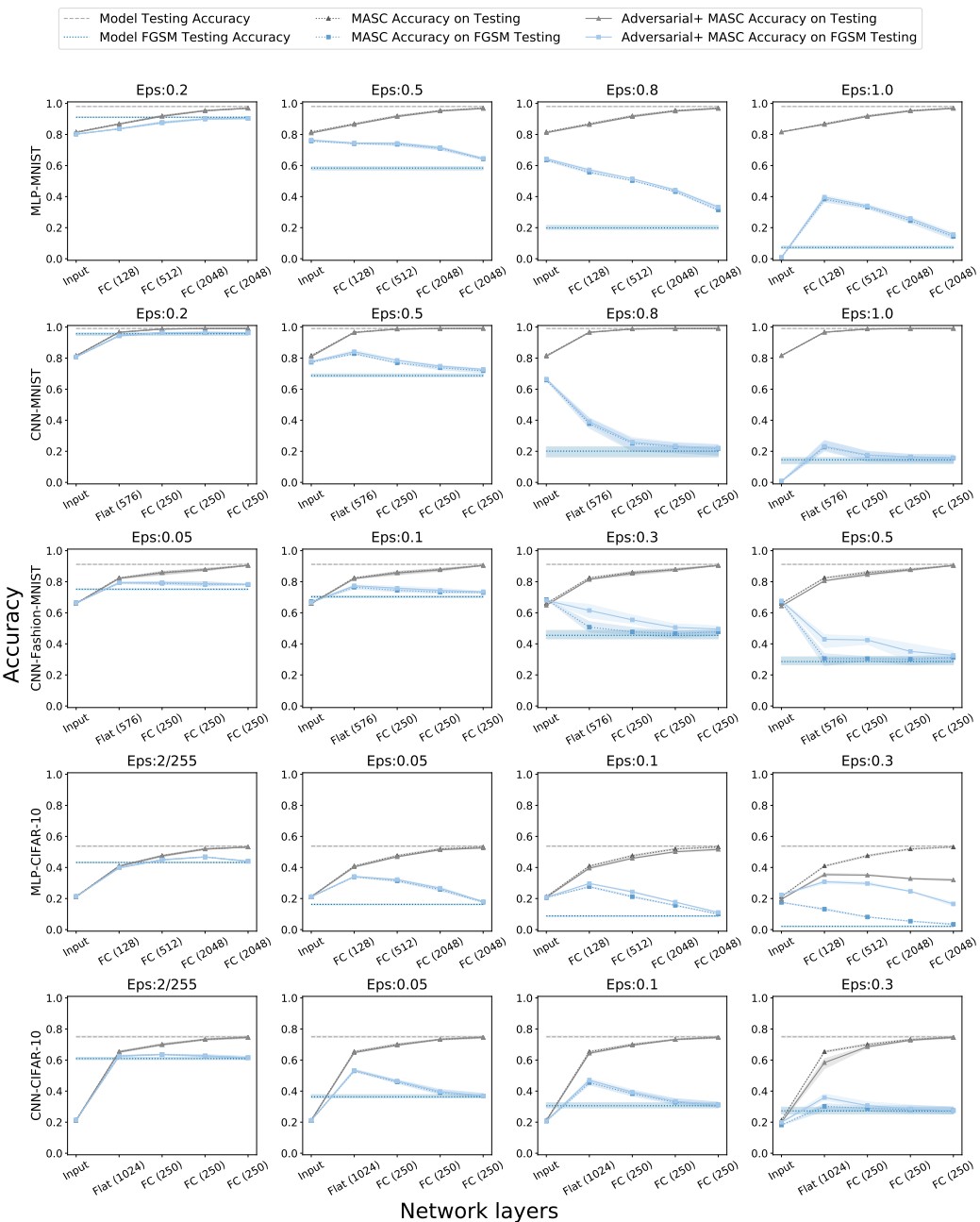

Figure 12: Adversarial+ Minimum Angle Subspace Classifier (Adversarial+ MASC) accuracy on adversarially perturbed test dataset and clean test dataset over the layers of the network and for varying $\epsilon$ values. Here, the data is projected onto class-specific subspaces constructed from FGSM and clean training dataset. $\epsilon$ value is presented at the top of each plot and the columns represent model-dataset pair as indicated. For reference, the model accuracy on clean test dataset and adversarially perturbed test dataset of the corresponding model (dotted line) is also shown. MASC accuracy on testing (dotted line) and FGSM testing (dotted line) when data is projected onto subspaces corresponding to only clean data is overlaid for comparison.

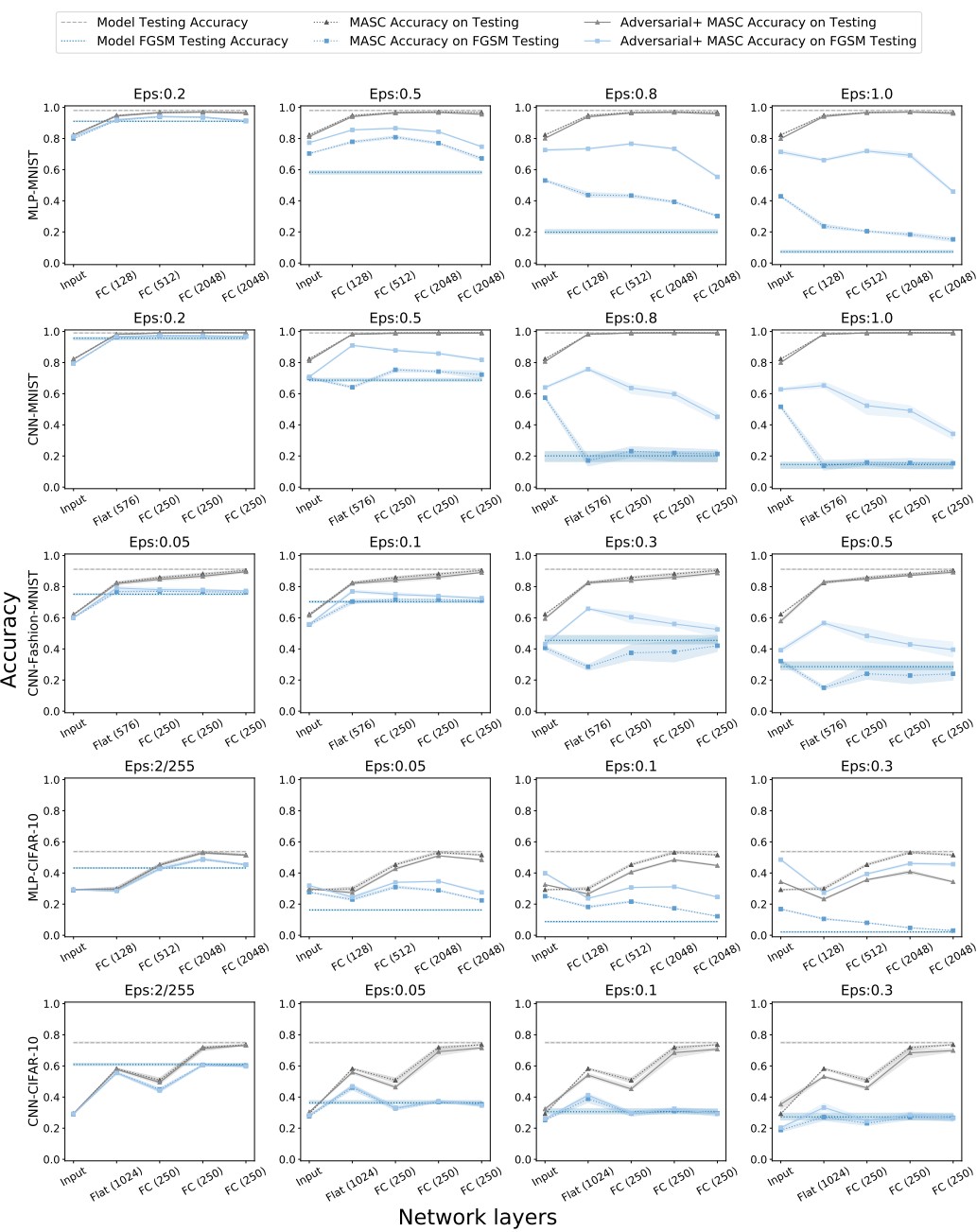

Figure 13: Adversarial+ Minimum Angle Subspace Classifier (Adversarial+ MASC) accuracy on adversarially perturbed test dataset and clean test dataset over the layers of the network and for varying $\epsilon$ values. Here, the data is projected onto class-specific subspaces constructed from FGSM and clean training dataset. $\epsilon$ value is presented at the top of each plot and the columns represent model-dataset pair as indicated. For reference, the model accuracy on clean test dataset and adversarially perturbed test dataset of the corresponding model (dotted line) is also shown. MASC accuracy on testing (dotted line) and FGSM testing (dotted line) when data is projected onto subspaces corresponding to only clean data is overlaid for comparison.

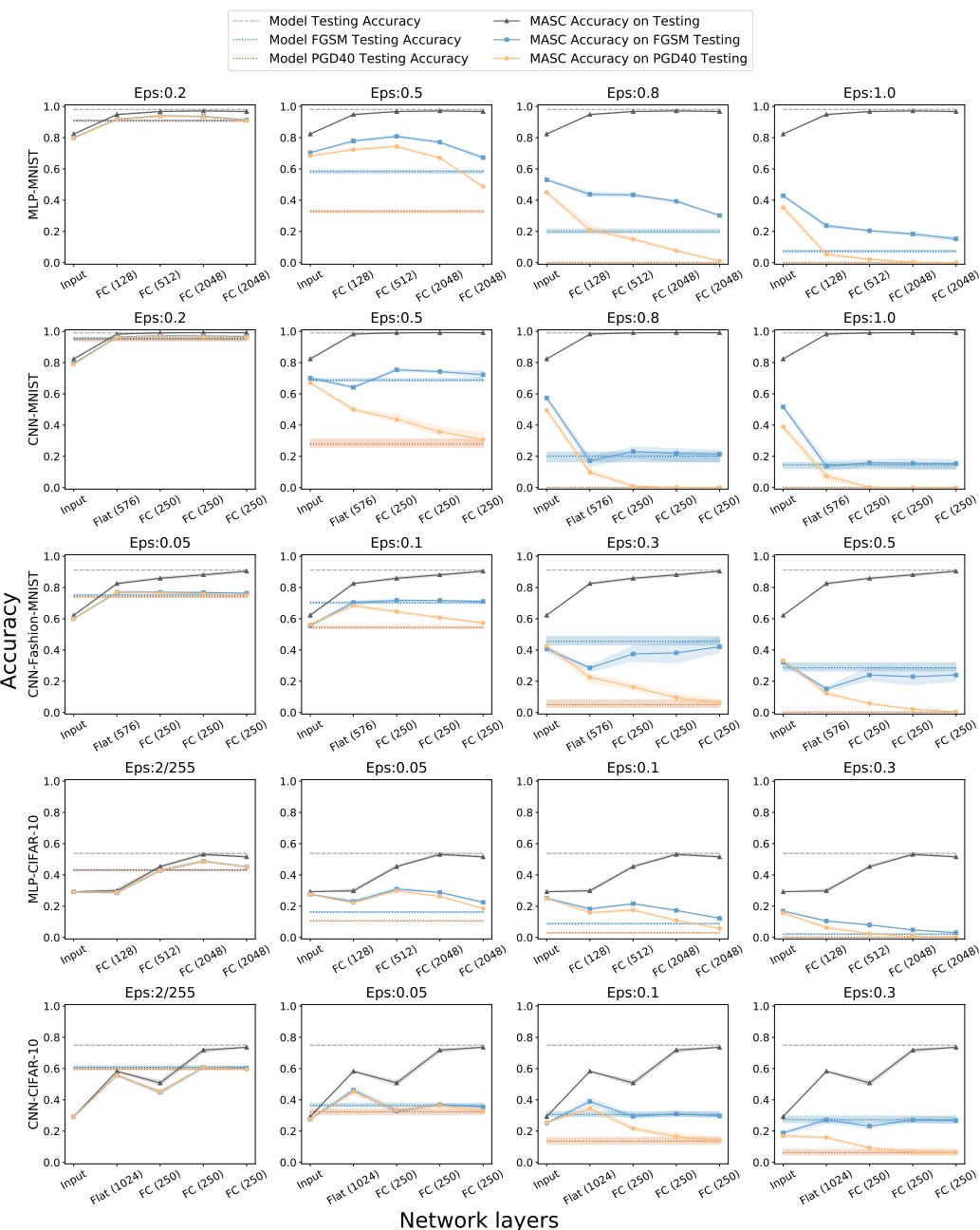

Figure 14: Minimum Angle Subspace Classifier (MASC) accuracy on adversarially perturbed test dataset and clean test dataset over the layers of the network and for varying $\epsilon$ values. Here, the data is projected onto class-specific subspaces constructed from the clean training dataset. $\epsilon$ value is presented at the top of each subplot and the rows represent model-dataset pair as indicated. For reference, the model accuracy on clean test dataset and adversarially perturbed test dataset of the corresponding model (dotted line) is also shown. PGD40 refers to Projected Gradient Descent (PGD) adversarial attacks run for 40 iterations (steps).

