# OpenReview forum: "Adversarially-robust probes for Deep Networks"
_NeurIPS.cc/2025/Workshop/Reliable_ML — NeurIPS 2025 - Reliable ML Workshop_

### Official Review · Reviewer_kCxY · 2025-09-17
**Promising Approach to Adversarial Robustness with MASC**

**Rating:** 8
**Confidence:** 4

**Review:**

Summary:
This paper concerns model robustness to adversarial attacks by introducing probes, based on the training data, at each layer in the model to classify based on learned representations of the data. The authors apply the Minimum Angle Subspace Classifier (MASC), a method known to improve generalization, on adversarial training data to improve robustness to attacks. The method was tested on several datasets and showed improvements over the base model for both clean and adversarial test data.

Strengths:
This is a comprehensive study that applies and expands on the MASC approach to adversarial data. The results show significant improvements in handling adversarial attacks compared to the baseline model. I found the approach both interesting and promising, particularly in how it leverages layer-wise probes to diagnose and improve robustness.

Weaknesses:
The presentation of the methods and results could be improved. Some important results are not clearly emphasized, and it is often difficult to tell from the figures what the reader should focus on or which method performs better or worse.

Suggestions:
A diagram could help clarify the explanation of the method (particularly how MASC works) potentially replacing parts of the textual explanation and making it more intuitive. Another diagram or schematic to organize the test cases and results would also help readers follow the experiments. Finally, the key results should be highlighted better visually so that improvements stand out clearly.

Ethics: N/A

---

### Official Review · Reviewer_amxo · 2025-09-19
**Exciting geometrical method to explore post-hoc probes on hidden representations of pre-trained networks.**

**Rating:** 8
**Confidence:** 1

**Review:**

The authors suggest that hidden layer representations in neural networks contain useful structure. These structures can be tapped into for adversarial robustness. They use the Minimum Angle Subspace Classifier to projects hidden activations into class-specific subspaces (with PCA) and assigns labels based on the closest angle. They test with and without adversarial data. They show that these probes often outperform the base model on adversarially perturbed inputs. In the ImageNet case, they report a ~3.5× improvement in adversarial accuracy.

Strengths

The paper has potential to understand what structure hidden layers really capture. This method is computationally cheap relative to adversarial training and performs just as well if not better.

Weaknesses

The choice of choosing which layer to probe is not clearly stated. The results imply that robustness varies across layers, but the paper leaves this as an observation rather than offering a systematic analysis. More examples and discussion would help understand the generalizability of the method.
The could benefit from theoretical justifications of why projecting onto class-conditioned subspaces improves adversarial robustness could help the reader understand the strength of this paper.